# Energy-Efficient Data Transmission for Underwater Wireless Sensor Networks: A Novel Hierarchical Underwater Wireless Sensor Transmission Framework

**DOI:** 10.3390/s23125759

**Published:** 2023-06-20

**Authors:** Jiasen Zhang, Xiaomei Wang, Bin Wang, Weikai Sun, Haiyang Du, Yuanke Zhao

**Affiliations:** 1School of Cyber Science and Engineering, Zhengzhou University, Zhengzhou 450002, China; zhangjiasen997@gs.zzu.edu.cn (J.Z.);; 2Information System Engineering College, PLA Strategic Support Force Information Engineering University, Zhengzhou 450001, China; oceansgroup2020@163.com (B.W.);

**Keywords:** UWSNs, energy-efficient data transmission, economic game theory, NIP problem, E-DDTMD

## Abstract

The complexity of the underwater environment enables significant energy consumption of sensor nodes for communication with base stations in underwater wireless sensor networks (UWSNs), and the energy consumption of nodes in different water depths is unbalanced. How to improve the energy efficiency of sensor nodes and meanwhile balance the energy consumption of nodes in different water depths in UWSNs are thus urgent concerns. Therefore, in this paper, we first propose a novel hierarchical underwater wireless sensor transmission (HUWST) framework. We then propose a game-based, energy-efficient underwater communication mechanism in the presented HUWST. It improves the energy efficiency of the underwater sensors personalized according to the various water depth layers of sensor locations. In particular, we integrate the economic game theory in our mechanism to trade off variations in communication energy consumption due to sensors in different water depth layers. Mathematically, the optimal mechanism is formulated as a complex nonlinear integer programming (NIP) problem. A new energy-efficient distributed data transmission mode decision algorithm (E-DDTMD) based on the alternating direction method of multipliers (ADMM) is thus further proposed to tackle this sophisticated NIP problem. The systematic simulation results demonstrate the effectiveness of our mechanism in improving the energy efficiency of UWSNs. Moreover, our presented E-DDTMD algorithm achieves significantly superior performance to the baseline schemes.

## 1. Introduction

In total, 71% of the Earth’s surface is covered by oceans, which are extremely rich in resources. With the curiosity of human beings for the unknown space and the growing demand for natural resources, the scientific exploration of the ocean has attracted more and more attention from researchers. Among them, the underwater wireless sensor network (UWSN) has great potential in the scientific exploration of the ocean through wireless sensing, collection, and real-time transmission of data [1]. Therefore, in recent years, UWSNs have been used to collect various underwater data to achieve many goals (such as pollution monitoring, disaster prevention, flood monitoring, oceanographic research, oil pipeline monitoring, mineral exploration, military surveillance, etc.) [2,3].

However, due to the particularity of the underwater environment, the underwater communication paradigm is different from the terrestrial communication paradigm. For instance, the radio frequency (RF) has a large attenuation in cases employed for underwater transmission. The RF can only be transmitted over short distances underwater and is, therefore, not suitable for underwater transmission [4], even though there has been some research on this [5,6,7,8]. As an alternative, the sound wave is commonly used to assist underwater communication. Therefore, UWSN technologies are implemented and deployed deep underwater with sensors using acoustic signals to perform communication [9]. Nevertheless, due to the narrow available bandwidth and high channel noise of underwater acoustic channels, underwater inter-sensor communication still requires significant power consumption [10,11]. Meanwhile, since the underwater acoustic sensors are energy-limited, the substantial communication energy consumption contributes to their service lifetime reduction [12]. Therefore, how to improve underwater acoustic sensors’ communication energy efficiency and extend the service life of sensor nodes are urgent challenges for UWSMs.

Some researchers effectively reduce the energy consumption of UWSNs by using an autonomous underwater vehicle (AUV) to assist data acquisition [13,14]. Yet, the delay for this data acquisition and processing scheme is excessive for some data with heightened importance. For transmission tasks that have requirements on transmission and the processing delay, they can only be transmitted to the BS through sensor nodes or sinking nodes. Therefore, other researchers have proposed some effective underwater data compression schemes, as well as energy-efficient UWSN routing protocols [15,16,17]. These reduce the energy consumption of underwater nodes to a certain extent and extend the service life of the UWSN.

Existing studies have improved the energy efficiency of nodes in UWSN to some extent. To the best of our knowledge, few existing studies, however, considered an essential concern, i.e., the communication energy consumption of nodes located in different water depths varies in UWSNs. This is because transmission energy consumption is different due to the different transmission paths of various nodes to ensure the quality of service (QoS). Therefore, if the influence of water depth is not considered, the existing research can reduce the overall energy consumption of underwater sensor networks. Yet, some sensors are potentially more energy-hungry. As a result, nodes need to be replaced frequently in different time periods, resulting in long-term service interruption.

This paper thus aims to improve the energy consumption efficiency of underwater node communication, while trading off the energy consumption of each node’s fairness. The main contributions of this paper are as follows:(1)We first propose a novel hierarchical underwater wireless sensor transmission (HUWST) framework. In the presented HUWST framework, sensor nodes are managed in clusters. The sink node (SN) acts as the cluster head; it converges the data collected by the whole cluster sensor nodes. Particularly, the SN has some computational capabilities and can assist the BS to process the important tasks to be transmitted in advance, as in [15]. In addition, the SN enables transmitting data to the BS or the upper SN directly for extremely important data.(2)We then propose a game-based, energy-efficient underwater communication mechanism based on the presented HUWST framework. We integrate the economic game theory in our mechanism to trade off variations in communication energy consumption due to sensors in different water depth layers.(3)In addition, we formulate the optimal mechanism as a nonlinear integer programming (NIP) problem, while satisfying the data volume constraint of the upper SN. We skillfully transform such a sophisticated NIP problem into a linear programming problem by means of a binary variable relaxation and decompose the problem. An energy-efficient distributed data transmission mode decision algorithm (E-DDTMD) based on the alternating direction method of multipliers (ADMM) is further proposed to tackle this complicated NIP problem.

The rest of this paper is organized as follows. In Section 2, we present related works and elaborate on the contributions of existing work and analyze their weaknesses. In Section 3, we have further statements with explanations about the applicability and practicality of our proposal. In Section 4, we describe the network model and propose the novel hierarchical underwater wireless sensor transmission (HUWST) framework. In Section 5, we formulate the computation and communication overhead of SN. In particular, based on economic game theory, we propose a payment mechanism. We then propose a game-based, energy-efficient underwater communication mechanism in the presented HUWST. In Section 6, a new E-DDTMD is thus further proposed to tackle this sophisticated NIP problem. The effectiveness of the proposed scheme is demonstrated by simulation results with different system parameters in Section 7.

## 2. Related Work

The importance of communication energy consumption of underwater sensor nodes has been highlighted in several research papers:

Zhou et al. [13] introduced an autonomous underwater vehicle (AUV)-aided underwater acoustic sensor network (UWSN). It utilized AUVs to assist data collection in UWSNs to reduce the number of data collection tasks of sensor nodes, thereby reducing the energy consumption of UWSN. Similarly, in [14], Khan et al. proposed an AUVs-assisted energy-efficient clustering (AEC) mechanism. It introduced the wake-up sleep cycle for the underwater nodes. Although [13,14] can effectively reduce the energy consumption of a UWSN, the delay for this data acquisition and processing scheme is excessive for some data with heightened importance. For transmission tasks that have requirements on transmission and the processing delay, they can only be transmitted to the BS through sensor nodes or sinking nodes. 

Therefore, Hu et al. [15] proposed two novel lightweight sensor data compression algorithms, i.e., the lossy data compression algorithm (LCA) and the lossless data compression algorithm (NLCA). In these algorithms, the sensors can transmit a smaller amount of data, reducing the energy consumption of sensor nodes. An energy-efficient routing protocol for selecting relay nodes in UWSMs based on the fuzzy analytical hierarchy process was presented in [16]. To some extent, the average energy consumption of each node in the UWSN receiving and transmitting data was reduced. Meanwhile, Zhang et al. [17] proposed energy-efficient probabilistic depth-based routing (EEPDBR) for UWSNs, which is improved from the traditional depth-based routing (DBR) algorithm. It takes a node’s depth, residual energy, and forwarding number within its 2-hop neighborhood into account. The algorithm can achieve a higher packet delivery ratio and lower average delivery time, while saving energy consumption effectively. The studies in [15,16,17] reduced the energy consumption of underwater nodes to a certain extent and extend the service life of the UWSN. 

Existing studies have improved the energy efficiency of nodes in a UWSN to some extent. To the best of our knowledge, few existing studies, however, considered an essential concern, i.e., the communication energy consumption of nodes located in different water depths varies in UWSNs. This is because the transmission energy consumption is different due to the different transmission paths of various nodes to ensure the quality of service (QoS). Therefore, if the influence of water depth is not considered, the existing research can reduce the overall energy consumption of underwater sensor networks. Yet, some sensors are potentially more energy-hungry. As a result, nodes need to be replaced frequently in different time periods, resulting in long-term service interruption. 

## 3. Applicability and Practicability Analysis of Proposal 

In this paper, we mainly consider that there are two significant usability concerns of our proposed hierarchical underwater wireless sensor transmission (HUWST) framework: (1) the ability of the underwater node to transmit data directly to the surface base station; (2) the computing capability of the underwater node to support our proposal.

In traditional underwater wireless sensor networks (UWSNs), the sensors are enabled to collect data and transmit the collected data to the surface base station (BS) [18]. This is due to the multi-hop forwarding mechanism between sensors [18]. The transmission distance of these sensors is, however, limited [19]. Underwater sensors in the deep are unable to communicate directly with the BS. Fortunately, the sink node (SN) with strong transmission capacity and computation capability has been deployed extensively in UWSNs [20,21]. Each SN is responsible for an underwater area, and sensors in this area allow transmitting data to the SN. The SN and sensors in this area form a cluster, with SN as the cluster head. SNs are not only able to transmit data to BSs via multi-hop forwarding between SNs, but also directly to BSs due to their robust transmission capability [22,23].

Therefore, based on real applicability and practicality, in this paper, we investigated the challenges faced by SN in UWSNs. Because the SN’s energy is not infinite and difficult to charge, similar to the sensor node, it also faces the problem of large energy consumption and imbalance. As for the sensors, the same as previously mentioned, the sensors only need to transmit the data to the SNs for further processing. The sensors’ energy consumption is thus assumed to be similar; we do not need to discuss the transmission problem of sensors. This paper, hence, takes SNs in the framework as the minimum analysis unit. Further, SNs have sufficient transmission capacity and computing power to support our proposal.

## 4. System Model

We first present the designed HUWST framework in Section 4.1. Subsequently, the communication model and the SN local computation model of the HUWST are introduced in Section 4.2 and Section 4.3, respectively.

### 4.1. HUWST

There are three types of entities in the considered HUWST framework (Figure 1). They respectively are the base station (BS), sink node (SN), and sensor. Among them, several sensors and an SN form a cluster, where the SN acts as a cluster head, and the sensor acts as a cluster member. Further, the cluster head is responsible for all the sensors in the cluster. Each sensor belongs to only one cluster, and the sensors hand over the collected data to the cluster head SN. The cluster head SN transmits the data to the BS uniformly. We detail the differences of three types of entities as follows.

(1)Sensors: The sensors have data acquisition capability and data transmission capability. They are used to collect underwater data and transmit the collected data to the cluster head SN of their cluster.(2)SNs: The SNs are used to receive the data transmitted by the sensors in their cluster and transmit the received data to the BS in a unified way. In particular, SN has certain computing power, which supports it to perform computational processing (e.g., data mining and analysis) on the received data.(3)BS: The BS can be a ship or surface monitoring station to receive data transmitted by underwater devices.

**Figure 1 sensors-23-05759-f001:**
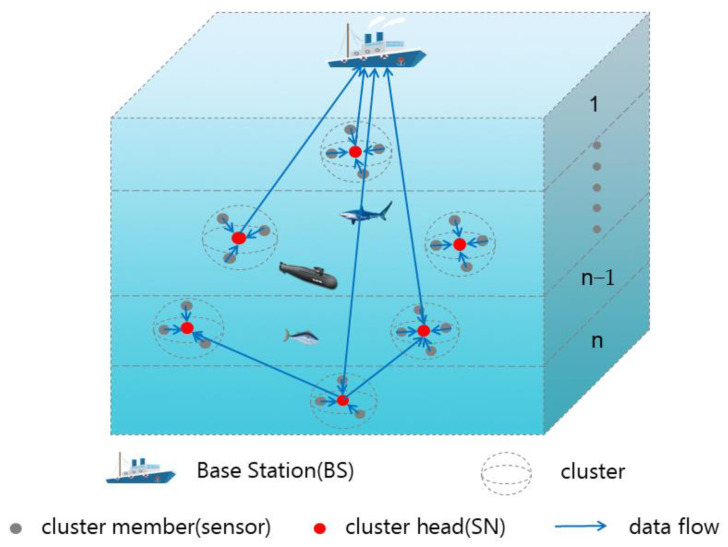
HUWST framework.

In the considered HUWST framework (Figure 1), we divide the network into n layers, in the vertical direction, each h meters as a layer. In HUWST, the SN has two choices for data transmission: ① Transmit data to the SNs of the upper layer (non-layer hopping transmission); ② Transmit data directly to the BS. To counterbalance the different energy consumption between different sensors, the sensor simply passes the task to the SN for further processing. The sensors’ energy consumption is thus assumed to be similar, so we do not discuss the transmission problem of sensors. The SN transmission model of the considered HUWST framework is shown in Figure 2 (The black dots in Figure 2 represent the omitted layers in the middle).

Therefore, the data transmission paradigm of each SN can be divided into four modes, which are:Transmit data directly to the BS;Transmit to the SN of the upper layer, to handle the data task further by the upper-layer SN;Transmit the calculation result directly to BS after calculating;After calculating and processing, transmit the calculation result to the SN of the upper layer, then the SN of the upper layer transmits it to the BS.

Because SNs typically have low data rates and suffer from selective fading in both time and frequency domains, we consider our novel HUWST framework based on the orthogonal frequency division multiplexing (OFDM) scheme. The OFDM scheme has the advantages of resisting frequency selective attenuation and improving frequency band utilization. Therefore, underwater OFDM systems have already become an essential choice for UWSNs [24,25]. For the underwater OFDM system, the total bandwidth is divided into multiple orthogonal sub-channels, and each sub-channel can be assigned to at most one sensor node. Because different sensor nodes use different frequency bands to transmit signals, the same as in previous studies [24,25], the interference between nodes can be ignored accordingly.

We pay attention to one layer and call it the optimization layer. We assume there are I SNs in this layer and use $I=\left\{1,2,3,\ldots ,I\right\}$ to denote them. The layer above this layer is called the assisted layer, and it is denoted that there are J SNs in the assisted layer, using $J=\left\{1,2,3,\ldots J\right\}$ to denote them. For each SN in the optimization layer, we consider that there is only one piece of data to be transmitted, and this piece of data cannot be divided. We use $W_{i}=\left\{D_{i},X_{i}\right\},i\in I$ to represent the data of SN i, where $D_{i}$ represents the data size, and $X_{i}$ represents the number of CPU cycles required to process it. The considered three-layer model—the optimization layer, the assisted layer, and the water surface BS—is shown in Figure 3 below. Table 1 summarizes the main notations used in this paper.

### 4.2. Communication Model

Unlike terrestrial communication networks, in underwater communication networks, underwater acoustic signals experience spreading and absorption losses, which are related to the signal frequency f. The total attenuation of the underwater acoustic signal over distance l can be denoted by [10]
(1)$$A\left(l,f\right)=l^{k}a{\left(f\right)}^{l},$$
where k represents the propagation geometry, and its usual value is 1.5. For f in kHz, the absorption coefficient $a\left(f\right)$ in dB/km is given by Thorp’s formula; we have [26]
(2)$$10\log a\left(f\right)=\frac{0.11f^{2}}{1+f^{2}}+\frac{44f^{2}}{4100+f^{2}}+2.75\times 10^{-4}f^{2}+0.003.$$

Based on this, when the signal frequency f, the signal bandwidth B, and the transmission power P of the underwater acoustic signal are given, the transmission rate R of data transmitted by the underwater communication device to the device with a distance of l from it can be expressed by
(3)$$R\left(B,P,l,f\right)=B\log_{2}(1+\frac{P}{A\left(l,f\right)N\left(f\right)B}),$$

The transmission delay of data with data volume D can be expressed as
(4)$$T=\frac{l}{c}+\frac{D}{R\left(B,P,l,f\right)},$$
where the symbol c represents the velocity of underwater sound, the power spectral density $N\left(f\right)$ of ambient noise in the ocean has four components; details of $N\left(f\right)$ can be found in [27]. The different types of power spectral density (p.s.d) of noisy sources components in dB are μ Pa per Hz and can be obtained as follows:(5a)$$10\log N_{t}\left(f\right)=17-30\log f,$$
(5b)$$10\log N_{s}\left(f\right)=40+20\left(s-\frac{1}{2}\right)+26\log f-60\log\left(f+0.03\right),$$
(5c)$$10\log N_{w}\left(f\right)=50+7.5w^{\frac{1}{2}}+20\log f-40\log\left(f+0.4\right),$$
(5d)$$10\log N_{th}\left(f\right)=-15+20\log f,$$
where f denotes the transmitting frequency. The combined noise $N\left(f\right)$ in the acoustic channel yields
(6)$$N\left(f\right)=N_{t}\left(f\right)+N_{s}\left(f\right)+N_{w}\left(f\right)+N_{th}\left(f\right).$$

### 4.3. SN Local Computation Model

For the optimization layer SN i, we use $f_{i}^{L}$ to denote its computing power; then, the energy consumption required by SN i to compute data task can be denoted by
(7)$$E_{i}^{C}=\varepsilon {\left(f_{i}\right)}^{2}X_{i},$$

The corresponding computation delay can be expressed as
(8)$$T_{i}^{L}=\frac{X_{i}}{f_{i}},$$
where ε denotes the energy factor, whose value depends on the coefficient of the chip architecture used [28]. We denote the data size after the computation as $D_{i}^{C}$.

## 5. Game-Based Energy-Efficient Underwater Communication Mechanism and NIP Problem 

In Section 5.1, we first propose a payment mechanism in conjunction with economic game theory. In Section 5.2, we calculate the overhead of SN i under four different transmission modes. In Section 5.3, we propose a game-based, energy-efficient underwater communication mechanism and formulate the optimal mechanism as a complex NIP problem.

### 5.1. Payment Mechanism

We propose a payment mechanism in conjunction with economic game theory. When the optimization layer SN i transmits data to the assisted layer SN j, SN j charges SN i a corresponding fee. The cost formula can be written as
(9)$$Q_{i,j}=k_{j}D_{i},$$

In addition, based on a non-cooperative game, we reasonably speculate that the total delay for SN j to further process data and transfer it to BS is:(10)$$T_{j,B}=t_{j}(\frac{l_{j,B}}{c}+\frac{D}{R\left(B_{j},P_{j,B},l_{j,B},f_{j}\right)}),$$
where $D_{i}$ is the data size transmitted from SN i to SN j. $k_{j}$ is the charging rate of SN j, i.e., the fee to be paid to SN j for transmitting the data of each data unit to SN j. In order to ensure unity with other overhead units, we set the $k_{j}$ unit as mJ/kB. Let $t_{j}$ be the monetary value of the non-cooperative game. We set the value of $k_{j}$ to be inversely proportional to the residual energy of SN j. When SN j has less residual energy, setting a higher charging rate can increase the cost.

### 5.2. Overhead of SN i

We incorporate the above payment mechanism into the overhead calculation. Then, we calculate the overhead of the four modes of SN i:

(1)Total overhead $C_{i}^{a}$ by mode *a*:The energy consumption of SN i to transmit data directly to BS can be expressed as
(11)$$E_{i}^{S}=P_{i,B}\frac{D_{i}}{R\left(B_{i},P_{i,B},l_{i,B},f_{i}\right)},$$
where $P_{i,B}$ is the transmit power of SN i to transmit data to BS, $l_{i,B}$ is the distance from SN i to BS, and $B_{i}$ and $f_{i}$ are the channel bandwidth and carrier frequency.

Total delay $T_{i}^{a}$ in the mode *a*:(12)$$T_{i}^{a}=\frac{l_{i,B}}{c}+\frac{D_{i}}{R\left(B_{i},P_{i,B},l_{i,B},f_{i}\right)},$$
the total overhead $C_{i}^{a}$ of SN i in mode a can be written as
(13)$$C_{i}^{a}=E_{i}^{S}.$$

(2)Total overhead $C_{i,j}^{b}$ by mode *b*:The energy consumption of SN i to transmit data to SN j can be denoted as
(14)$$E_{i,j}^{S}=P_{i,j}\frac{D_{i}}{R\left(B_{i},P_{i,j},l_{i,j},f_{i}\right)},$$
where $P_{i,j}$ is the transmit power of SN i to transmit data to SN j, and $l_{i,j}$ is the distance from SN i to SN j.

Total delay $T_{i,j}^{b}$ in the mode *b*:(15)$$T_{i,j}^{b}=\frac{l_{i,j}}{c}+\frac{D_{i}}{R\left(B_{i},P_{i,j},l_{i,j},f_{i}\right)}+T_{j,B}^{b}.$$

The total overhead $C_{i,j}^{b}$ of SN i in mode b can be written as
(16)$$C_{i,j}^{b}=E_{i,j}^{S}+Q_{i,j}.$$

(3)Total overhead $C_{i}^{c}$ by mode *c*:The energy consumption for SN i to transmit the calculation result to BS can be expressed as
(17)$$E_{i}^{C,S}=P_{i,B}\frac{D_{i}^{C}}{R\left(B_{i},P_{i,B},l_{i,B},f_{i}\right)},$$
where $D_{i}^{c}$ is the amount of data to calculate the processed result.

Total delay $T_{i}^{c}$ in the mode *c*:(18)$$T_{i}^{c}=T_{i}^{L}+\frac{l_{i,B}}{c}+\frac{D_{i}^{C}}{R\left(B_{i},P_{i,B},l_{i,B},f_{i}\right)},$$

The total overhead $C_{i}^{c}$ of SNi in mode c can be denoted by
(19)$$C_{i}^{c}=E_{i}^{C}+E_{i}^{C,S}.$$

(4)Total overhead $C_{i,j}^{d}$ by mode *d*:The energy consumption for SN i to transmit the calculation result to SN j can be expressed as
(20)$$E_{i}^{C,S}=P_{i,j}\frac{D_{i}^{C}}{R\left(B_{i},P_{i,j},l_{i,j},f_{i}\right)}.$$

Total delay $T_{i,j}^{d}$ in the mode d:(21)$$T_{i,j}^{d}=T_{i}^{L}+\frac{l_{i,j}}{c}+\frac{D_{i}^{C}}{R\left(B_{i},P_{i,j},l_{i,j},f_{i}\right)}+T_{j,B}^{d},$$
where $T_{j,B}^{d}$ is represented by equation $T_{j,B}^{d}=t_{j}(\frac{l_{j,B}}{c}+\frac{D_{i}^{C}}{R\left(B_{j},P_{j,B},l_{j,B},f_{i}\right)})$.

The total overhead $C_{i,j}^{d}$ of SN i in mode *d* can be written as
(22)$$C_{i,j}^{d}=E_{i}^{C}+E_{i,j}^{C,S}+k_{j}D_{i}^{C}.$$

### 5.3. The Optimal Mechanism and Problem Formulation

Based on economic game theory, we propose a payment mechanism to incorporate overhead. Therefore, the game-based, energy-efficient underwater communication mechanism can be expressed as minimizing the total overhead of the optimization layer SNs.

We use $a_{i}\in \left\{0,1\right\}$ to denote whether SN i chooses mode *a*, i.e., transmits data directly to the BS, where $a_{i}=1$ means that mode *a* is selected, otherwise it is not selected, and the corresponding strategy can be represented by $a={\left\{a_{i}\right\}}_{i\in I}$. We use $b_{i,j}\in \left\{0,1\right\}$ to denote whether SN i chooses mode *b* and transmits data to SN j for its assistance, where $b_{i,j}=1$ means that SN i chooses mode *b*, otherwise it does not choose. The corresponding strategy can be represented by $b={\left\{b_{i,j}\right\}}_{i\in I,j\in J}$. Similarly, we use $d_{i,j}\in \left\{0,1\right\}$ to denote whether SN i chooses mode *d* to process the data task, i.e., assisted by SN j after calculating. As above, $d_{i,j}=1$ means selection, otherwise it means no selection, and the corresponding strategy can be represented by $d={\left\{d_{i,j}\right\}}_{i\in I,j\in J}$.

Since the optimization layer has only one datum per SN and cannot be split, we have the constraint as
(23)$$\sum_{j=1}^{J}\left(b_{i,j}+d_{i,j}\right)+a_{i}\leq 1,i\in \left\{1,2,\ldots I\right\}.$$

Therefore, the total overhead of all SNs in the optimization layer can be denoted as
(24)$$\overset{I}{\underset{i=1}{\sum }}\{a_{i}C_{i}^{a}+\overset{J}{\underset{j=1}{\sum }}\left(b_{i,j}C_{i,j}^{b}+d_{i,j}C_{i,j}^{d}\right)+[1-a_{i}-\overset{J}{\underset{j=1}{\sum }}\left(b_{i,j}+d_{i,j}\right)]C_{i}^{c}\}$$

It can be simplified as
(25)$$\overset{I}{\underset{i=1}{\sum }}\{\overset{J}{\underset{j=1}{\sum }}[b_{i,j}\left(C_{i,j}^{b}-C_{i}^{c}\right)+d_{i,j}\left(C_{i,j}^{d}-C_{i}^{c}\right)+a_{i}\left(C_{i}^{a}-C_{i}^{c}\right)+C_{i}^{c}]\}$$

At the same time, since the data size accepted by each SN in the assisted layer is limited in each round of task processing, we have the constraints as
(26)$$\sum_{i=1}^{I}\left(b_{i,j}D_{i}+d_{i,j}D_{i}^{C}\right)\leq Z_{j},j\in \left\{1,2,\ldots ,J\right\},$$
where $Z_{j}$ is the maximum data size that can be accepted by the assisted layer SN j during a single round of transmission. 

We set the following delay constraints to ensure its efficiency in terms of transmission time.
(27)$$\sum_{j=1}^{J}\left[b_{i,j}\left(T_{i,j}^{b}-T_{i}^{c}\right)+d_{i,j}\left(T_{i,j}^{d}-T_{i}^{c}\right)\right]+a_{i}\left(T_{i}^{a}-T_{i}^{c}\right)+T_{i}^{c}\leq T_{i},\forall i,$$
where $T_{i}$ is the maximum delay acceptable for transmitting SN i data to BS.

Apparently, with satisfying the limit of the number of accepted tasks per round for each SN in the upper layer and the delay constraint of SN i, the optimal mechanism can be written as
(28a)$$\underset{a,b,d}{min}\sum_{i=1}^{I}\left\{\sum_{j=1}^{J}\left[b_{i,j}\left(C_{i,j}^{b}-C_{i}^{c}\right)+d_{i,j}\left(C_{i,j}^{d}-C_{i}^{c}\right)\right]+a_{i}\left(C_{i}^{a}-C_{i}^{c}\right)+C_{i}^{c}\right\},$$
(28b)$$s.t.\sum_{i=1}^{I}\left(b_{i,j}D_{i}+d_{i,j}D_{i}^{C}\right)\leq Z_{j},j\in \left\{1,2,\ldots ,J\right\},$$
(28c)$$\sum_{j=1}^{J}\left(b_{i,j}+d_{i,j}\right)+a_{i}\leq 1,i\in \left\{1,2,\ldots I\right\},$$
(28d)$$\sum_{j=1}^{J}\left[b_{i,j}\left(T_{i,j}^{b}-T_{i}^{c}\right)+d_{i,j}\left(T_{i,j}^{d}-T_{i}^{c}\right)\right]+a_{i}\left(T_{i}^{a}-T_{i}^{c}\right)+T_{i}^{c}\leq T_{i},\forall i,$$
(28e)$$a_{i},b_{i,j},d_{i,j}\in \left\{0,1\right\},\forall i,j.$$

The objective function (28a) aims to minimize the total overhead of SN in the optimization layer. The constraint (28b) ensures that the total data size transmitted by all SNs in the optimization layer to SN j in each round of transmission does not exceed its maximum acceptable data size. The constraint condition (28c) indicates that the optimization layer adopts only one way per SN to transmit data. In addition, the constraint condition (28d) represents the delay constraint.

Since the objective function (28a) and constraints (28b)–(28e) are discrete and nonlinear, the optimization problem (28) is a complex nonlinear integer programming (NIP) problem [29]. To solve this problem, we transform the NIP problem into a linear programming (LP) problem through a binary variable relaxation method and decompose the problem in order to solve it in a distributed and efficient manner. Then, a new energy-efficient distributed data transmission mode decision algorithm(E-DDTMD) based on the alternating direction method of multipliers (ADMM) is thus further proposed to tackle this distributed problem.

## 6. E-DDTMD Algorithm Based on ADMM

In Section 6.1, we first transform the NIP problem (28) into a LP problem (29) by relaxing binary variables to continuous variables. In Section 6.2, to enable each SN in the optimization layer to participate in the problem-solving computation, we separate problem (29). In Section 6.3, we propose a new E-DDTMD based on ADMM to tackle this distributed problem. Then, we restore the continuous variables to binary variables by algorithm.

### 6.1. Problem Transformation

Because the binary variables $a_{i}$, $b_{i,j},$ and $d_{i,j}$ are included in the objective function and constraints, problem (28) is discrete and nonconvex. To solve this problem efficiently, we relax the binary variables $a_{i}$, $b_{i,j},$ and $d_{i,j}$ to continuous variables $0\leq a_{i}\leq 1,\;0\leq b_{i,j}\leq 1$ and $0\leq d_{i,j}\leq 1$. Therefore, the primal problem in (28) can be written as
(29a)$$\underset{a,b,d}{min}\sum_{i=1}^{I}\{\sum_{j=1}^{J}\left[b_{i,j}\left(C_{i,j}^{b}-C_{i}^{c}\right)+d_{i,j}\left(C_{i,j}^{d}-C_{i}^{c}\right)\right]+a_{i}\left(C_{i}^{a}-C_{i}^{c}\right)+C_{i}^{c}\},$$
(29b)$$s.t.\sum_{i=1}^{I}\left(b_{i,j}D_{i}+d_{i,j}D_{i}^{C}\right)\leq Z_{j},j\in \left\{1,2,\ldots ,J\right\},$$
(29c)$$\sum_{j=1}^{J}\left(b_{i,j}+d_{i,j}\right)+a_{i}\leq 1,i\in \left\{1,2,\ldots I\right\},$$
(29d)$$\sum_{j=1}^{J}\left[b_{i,j}\left(T_{i,j}^{b}-T_{i}^{c}\right)+d_{i,j}\left(T_{i,j}^{d}-T_{i}^{c}\right)\right]+a_{i}\left(T_{i}^{a}-T_{i}^{c}\right)+T_{i}^{c}\leq T_{i},\forall i,$$
(29e)$$0\leq a_{i},b_{i,j},d_{i,j}\leq 1,\forall i,j.$$

It is observed that the transformed problem (29) is a linear programming (LP) problem.

### 6.2. Problem Decomposition

For the optimization layer to perform problem-solving computation locally for each SN, the problem needs to be separated. However, the variables b and d in problem (29) are global variables and cannot be separated. Therefore, to make the problem separable, we introduce local copies of the global variables b and d so that each SN can use its local copy to compute independently to solve the problem. There will be global variables b and d in the optimization of the local copy on layer SN i using ${\hat{b}}^{i}={\left\{{\hat{b}}_{k,j}^{i}\right\}}_{k\in I,j\in J}$ and ${\hat{d}}^{i}={\left\{{\hat{d}}_{k,j}^{i}\right\}}_{k\in I,j\in J}$.

Define the following set as the local variable feasible set on SN i:(30)$$\varphi_{i}=\;\left\{\begin{array}{c} a_{i}\\ {\hat{b}}^{i}\\ {\hat{d}}^{i} \end{array}\right.\left|\left.\begin{array}{c} \sum_{k=1}^{I}({\hat{b}}_{k,j}^{i}D_{k}+{\hat{d}}_{k,j}^{i}D_{k}^{C})\leq Z_{j},\forall j\\ \sum_{j=1}^{J}\left({\hat{b}}_{k,j}^{i}+{\hat{d}}_{k,j}^{i}\right)+a_{i}\leq 1,\forall k\\ \sum_{j=1}^{J}[{\hat{b}}_{k,j}^{i}\left(T_{i,j}^{b}-T_{i}^{c}\right)+{\hat{d}}_{k,j}^{i}\left(T_{i,j}^{d}-T_{i}^{c}\right)]+a_{i}\left(T_{i}^{a}-T_{i}^{c}\right)+T_{i}^{c}\leq T_{i},\forall k\\ 0\leq a_{i}\leq 1\\ 0\leq {\hat{b}}_{k,j}^{i}\leq 1\\ 0\leq {\hat{d}}_{k,j}^{i}\leq 1 \end{array}\right\}\right.,\forall i.$$

The local utility function on SN i can be denoted as
(31)$$v_{i}\left(a_{i},{\hat{b}}^{i},{\hat{d}}^{i}\right)\;=\left\{\begin{array}{c} \overset{J}{\underset{j=1}{\sum }}[{\hat{b}}_{k,j}^{i}\left(C_{i,j}^{b}-C_{i}^{c}\right)+{\hat{d}}_{k,j}^{i}\left(C_{i,j}^{d}-C_{i}^{c}\right)]+a_{i}\left(C_{i}^{a}-C_{i}^{c}\right)+C_{i}^{c},when\left(a_{i},{\hat{b}}^{i},{\hat{d}}^{i}\right)\in \varphi_{i}\\ 0.otherwise \end{array}\right.$$

Then the equivalent formulation of problem (29) can be expressed as [30].
(32a)$$\underset{\left\{a,{\hat{b}}^{i},{\hat{d}}^{i}\}\{b,d\right\}}{min}\sum_{i\in I}v_{i}(a_{i},{\hat{b}}^{i},{\hat{d}}^{i}),$$
$$S.t.{\hat{b}}_{k,j}^{i}=b_{i,j},\forall i,j,k,$$
(32b)$${\hat{d}}_{k,j}^{i}=d_{i,j},\forall i,j,k.$$

Clearly, in problem (32a), the objective function $v_{i}$ with feasible set $\varphi_{i}$ is separable with respect to all SNs in the optimization layer, and the separation of the objective function enables each small unit to deal with the subproblems related to it independently. Additionally, constraint (32b) ensures consistency between all local and global variables, which is exactly what we want. In Section 6.3, we will apply a new energy-efficient distributed data transmission mode decision algorithm (E-DDTMD) based on ADMM [31] to solve this problem in a distributed manner.

### 6.3. E-DDTMD Algorithm Based on ADMM

According to [32], we have the augmented Lagrangian function of problem (32) as
(33)$$\begin{array}{l} L_{\rho }({\left\{a_{i},{\hat{b}}^{i},{\hat{d}}^{i}\right\}}_{i\in I},\left\{b,d\right\},{\left\{\delta^{i},\sigma^{i}\right\}}_{i\in I})\\ =\underset{i\in I}{\sum }v_{i}(a_{i},{\hat{b}}^{i},{\hat{d}}^{i})\\ +\underset{i\in I}{\sum }\underset{k\in I,j\in J}{\sum }\delta_{k,j}^{i}({\hat{b}}_{k,j}^{i}-b_{k,j})+\underset{i\in I}{\sum }\underset{k\in I,j\in J}{\sum }\sigma_{k,j}^{i}({\hat{d}}_{k,j}^{i}-d_{k,j})\\ +\frac{\rho }{2}\sum_{i\in I}\sum_{k\in I,j\in J}\|{\hat{b}}_{k,j}^{i}-b_{k,j}{\|}^{2}+\frac{\rho }{2}\sum_{i\in I}\sum_{k\in I,j\in J}\|{\hat{d}}_{k,j}^{i}-d_{k,j}{\|}^{2}, \end{array}$$
where the $\sigma^{i}=\left\{\sigma_{k,j}^{i}\right\}$ and $\delta^{i}=\left\{\sigma_{k,j}^{i}\right\}$ and is problem (33) of the Lagrange multiplier, ρ is punish coefficient [30].

E-DDTMD based on ADMM used in through an iterative update ${\left\{a_{i},{\hat{b}}^{i},{\hat{d}}^{i}\right\}}_{i\in I}\;,\left\{b,d\right\}$ and ${\left\{\delta^{i},\sigma^{i}\right\}}_{i\in I}$ to solve (33), set ${\left\{a_{i},{\hat{b}}^{i},{\hat{d}}^{i}\right\}}_{i\in I}^{\left[t\right]},\;{\left\{b,d\right\}}^{\left[t\right]}$ and ${\left\{\delta^{i},\sigma^{i}\right\}}_{i\in I}^{\left[t\right]}$ corresponding to the first *t* iteration ${\left\{a_{i},{\hat{b}}^{i},{\hat{d}}^{i}\right\}}_{i\in I},\;\left\{b,d\right\}$ and ${\left\{\delta^{i},\sigma^{i}\right\}}_{i\in I}$. Below, according to the following steps, we update the (*t* + 1) iteration when ${\left\{a_{i},{\hat{b}}^{i},{\hat{d}}^{i}\right\}}_{i\in I},\left\{b,d\right\},$ and ${\left\{\delta^{i},\sigma^{i}\right\}}_{i\in I}$ value.

(1)SN local variables update ${\left\{a_{i},{\hat{b}}^{i},{\hat{d}}^{i}\right\}}_{i\in I}$ update:

In this step, we need to update the local variables. In a given ${\left\{b,d\right\}}^{\left[t\right]}$ and ${\left\{\delta^{i},\sigma^{i}\right\}}_{i\in I}^{\left[t\right]}$, the local variable ${\left\{a_{i},{\hat{b}}^{i},{\hat{d}}^{i}\right\}}_{i\in I}^{\left[t+1\right]}$ is updated; the equivalent to the following questions needs to be solved:(34)$${\left\{a_{i},{\hat{b}}^{i},{\hat{d}}^{i}\right\}}_{i\in I}^{\left[t+1\right]}=\arg\underset{\left\{a_{i},{\hat{b}}^{i},{\hat{d}}^{i}\right\}}{min}L_{\rho }({\left\{a_{i},{\hat{b}}^{i},{\hat{d}}^{i}\right\}}_{i\in I},{\left\{b,d\right\}}^{\left[t\right]},{\left\{\delta^{i},\sigma^{i}\right\}}_{i\in I}^{\left[t\right]}).$$

For problem (34), we can decompose it into I subproblems, each of which is solved locally and independently by each SN in the optimization layer. Therefore, for SN i, the following equivalent optimization problem is required to be solved when updating the local variables at iteration (*t* + 1):(35a)$$\underset{\left\{a_{i},{\hat{b}}_{k,j}^{i},{\hat{d}}_{k,j}^{i}\right\}}{min}\{v_{i}(a_{i},{\hat{b}}^{i},{\hat{d}}^{i})\;+\underset{k\in I,j\in J}{\sum }\delta_{k,j}^{i}({\hat{b}}_{k,j}^{i}-b_{k,j})+\underset{k\in I,j\in J}{\sum }\sigma_{k,j}^{i}({\hat{d}}_{k,j}^{i}-d_{k,j})\;+\frac{\rho }{2}\sum_{k\in I,j\in J}\|{\hat{b}}_{k,j}^{i}-b_{k,j}{\|}^{2}+\frac{\rho }{2}\sum_{k\in I,j\in J}\|{\hat{d}}_{k,j}^{i}-d_{k,j}{\|}^{2}\},$$
(35b)$$S.t.\{a_{i},{\hat{b}}^{i},{\hat{d}}^{i}\}\in \varphi_{i}.$$

Obviously, problem (35) is a convex problem, so the optimal solution can be obtained by using the primal-dual interior-point algorithm or the CVX toolkit [33].

(2)Update the $\left\{b,d\right\}$ global variables:

In this step, we need to update the global variables. Through the previous steps, we get the local variable ${\left\{a_{i},{\hat{b}}^{i},{\hat{d}}^{i}\right\}}_{i\in I}^{\left[t+1\right]}$; for a given ${\left\{a_{i},{\hat{b}}^{i},{\hat{d}}^{i}\right\}}_{i\in I}^{\left[t+1\right]}$, a global variable $\left\{b,d\right\}$ updated based on the following formula:(36)$${\left\{b\right\}}^{\left[t+1\right]}=\arg\underset{\left\{b_{k,j}\right\}}{min}[\sum_{i\in I}\sum_{k\in I,j\in J}\delta_{k,j}^{i\left[t\right]}({\hat{b}}_{k,j}^{i\left[t+1\right]}-b_{k,j})+\frac{\rho }{2}\sum_{i\in I}\sum_{k\in I,j\in J}\|{\hat{b}}_{k,j}^{i\left[t+1\right]}-b_{k,j}{\|}^{2}],$$
(37)$${\left\{d\right\}}^{\left[t+1\right]}=\arg\underset{\left\{b_{k,j}\right\}}{min}[\sum_{i\in I}\sum_{k\in I,j\in J}\sigma_{k,j}^{i\left[t\right]}({\hat{d}}_{k,j}^{i\left[t+1\right]}-d_{k,j})+\frac{\rho }{2}\sum_{i\in I}\sum_{k\in I,j\in J}\|{\hat{d}}_{k,j}^{i\left[t+1\right]}-d_{k,j}{\|}^{2}].$$

Since the above is an unconstrained quadratic convex problem, we can solve it by simply setting the gradient of $\left\{b\right\}$ and $\left\{d\right\}$ to zero, i.e.,
(38)$$\sum_{i\in I}\delta_{k,j}^{i\left[t\right]}+\rho \sum_{i\in I}({\hat{b}}_{k,j}^{i\left[t+1\right]}-b_{k,j})=0,\forall k,j,$$
(39)$$\sum_{i\in I}\sigma_{k,j}^{i\left[t\right]}+\rho \sum_{i\in I}({\hat{d}}_{k,j}^{i\left[t+1\right]}-d_{k,j})=0,\forall k,j,$$

This leads to
(40)$$b_{k,j}^{\left[t+1\right]}=\frac{1}{I\rho }\sum_{i\in I}\delta_{k,j}^{i\left[t\right]}+\frac{1}{I}\sum_{i\in I}{\hat{b}}_{k,j}^{i\left[t+1\right]},\forall k,j,$$
(41)$$d_{k,j}^{\left[t+1\right]}=\frac{1}{I\rho }\sum_{i\in I}\sigma_{k,j}^{i\left[t\right]}+\frac{1}{I}\sum_{i\in I}{\hat{d}}_{k,j}^{i\left[t+1\right]},\forall k,j.$$

By the Lagrange multiplier in the first *t* iteration initialized to zero, namely, the $\sum_{i\in I}\delta_{k,j}^{i\left[t\right]}=0$ and $\sum_{i\in I}\sigma_{k,j}^{i\left[t\right]}=0,\forall k,j$, we can make the type reduction for:(42)$$b_{k,j}^{\left[t+1\right]}=\frac{1}{I}\sum_{i\in I}{\hat{b}}_{k,j}^{i\left[t+1\right]},\forall k,j,$$
(43)$$d_{k,j}^{\left[t+1\right]}=\frac{1}{I}\sum_{i\in I}{\hat{d}}_{k,j}^{i\left[t+1\right]},\forall k,j.$$

(3)The Lagrange multiplier ${\left\{\delta^{i},\sigma^{i}\right\}}_{i\in I}$ is updated:

The Lagrange multiplier is updated according to the following formula:(44)$${\left\{\delta^{i}\right\}}_{i\in I}^{\left[t+1\right]}=\delta^{i\left[t\right]}+\rho ({\hat{b}}^{i\left[t+1\right]}-b_{i}^{\left[t+1\right]}),$$
(45)$${\left\{\sigma^{i}\right\}}_{i\in I}^{\left[t+1\right]}=\sigma^{i\left[t\right]}+\rho ({\hat{d}}^{i\left[t+1\right]}-d_{i}^{\left[t+1\right]}).$$

(4)Iteration stopping criterion: The original residual of each SN must be as small as possible under feasible conditions. Therefore, we set the iteration stopping threshold as

(46)$$\|{\hat{b}}^{i\left[t+1\right]}-b_{i}^{\left[t+1\right]}{\|}_{2}\leq \varepsilon_{pri},\forall i,$$(47)$$\|{\hat{d}}^{i\left[t+1\right]}-d_{i}^{\left[t+1\right]}{\|}_{2}\leq \varepsilon_{pri},\forall i,$$
where $\varepsilon_{pri}>0$ is the threshold in primal feasibility conditions. We set the dual residuals in feasibility conditions as
(48)$$\|b_{i}^{\left[t+1\right]}-b_{i}^{\left[t\right]}{\|}_{2}\leq \varepsilon_{dual},\forall i,$$
(49)$$\|d_{i}^{\left[t+1\right]}-d_{i}^{\left[t\right]}{\|}_{2}\leq \varepsilon_{dual},\forall i,$$
where $\varepsilon_{dual}>0$ is the threshold in dual feasibility conditions.

(5)Binary variable recovery:

Because we relaxed the binary variables to continuous variables in Section 6.1, we need to restore the continuous variables to binary variables by Algorithm 1 after reaching a feasible solution. We recover variables $b_{i,j}$ and $d_{i,j}$ by Algorithm 1 ($b_{i,j}$ as an example) [34]. For SN i, ∀i∈I, when ${\left\{b_{i,j}\right\}}_{j\in J}=0$ and ${\left\{d_{i,j}\right\}}_{j\in J}=0$, $a_{i}=0$ and $a_{i}=1$ respectively, into $v_{i}\left(a_{i},0,0\right)$ and recorded as $v_{i0}$ and $v_{i1}$; when $v_{i0}>v_{i1}$ return $a_{i}$ is 1; otherwise, $a_{i}$ is restored to 0. Moreover, the proposed energy-efficient distributed data transmission mode decision algorithm (E-DDTMD) based on ADMM is summarized in Algorithm 2.
**Algorithm 1: Binary Variables Recovery Algorithm.**Compute augmented Lagrangian’s partial derivations $Q_{i,j}=\partial L_{\rho }/\partial b_{i,j}$ for each $b_{i,j}$.Sort $Q_{i,j},\forall i,j$ from the largest to the smallest. Write them as $Q_{1},Q_{2}\ldots Q_{m}\ldots ,$ meanwhile write corresponding $b_{i,j}$ as $b_{1},b_{2}..b_{m}\ldots$**For** *m* = 1,2,…,**Do**
Set $b_{m}=1$ and $b_{m+1},b_{m+2},b_{m+3}\ldots =0$**If** Any of the constrains (28b)–(28e) does not hold, **Then Break**.**End for**4.Output the recovered binary variables $\left\{b_{i,j}\right\},\forall i,j$


**Algorithm 2: E-DDTMD Algorithm Based on ADMM.**

Initialization
(a)The iteration stopping threshold $\varepsilon_{pri}>0$ and $\varepsilon_{dual}>0$;(b)The initial solution ${\left\{a,b,d\right\}}^{\left[0\right]}$;(c)The initial Lagrange multipliers vectors ${\left\{\delta^{i\left[0\right]}>0,\sigma^{i\left[0\right]}>0\right\}}_{i\in I}$;
  *t* = 0.


2.Iterations
**Repeat**
(a)Each SN i∈I updates local variables ${\left\{a_{i},{\hat{b}}^{i},{\hat{d}}^{i}\right\}}_{i\in I}^{\left[t+1\right]}$;(b)Update the global variables ${\left\{b,d\right\}}^{\left[t+1\right]}$;(c)Update the Lagrange multipliers ${\left\{\delta^{i},\sigma^{i}\right\}}_{i\in I}^{\left[t+1\right]}$;
*t* = *t* + 1.
**Until**
$\|b_{i}^{\left[t+1\right]}-b_{i}^{\left[t\right]}{\|}_{2}\leq \varepsilon_{dual}$,$\|d_{i}^{\left[t+1\right]}-d_{i}^{\left[t\right]}{\|}_{2}\leq \varepsilon_{dual}$, ∀*i*
3.OutputOutput ${\left\{a,b,d\right\}}^{*}$ as optimal solution.


## 7. Simulation Results

In this section, we first evaluated the performance of our proposed E-DDTMD algorithm based on ADMM. Then, we simulated and compared the performance of the proposed optimal mechanism and the following two mechanisms. Finally, the systematic simulation results demonstrated our mechanism is effective in improving the energy efficiency of UWSNs, meanwhile balancing different depths SNs’ energy.

(1) Transmission after calculation (TAC): for TAC, each SN first computes and processes the data, and then transmits the results to the BS; 

(2) Direct transmission to BS (TDBS): For TDBS, each SN transmits data directly to the BS. 

We perform the simulation in MATLAB 2018b. In this simulation, there are J = 6 SNs in the assisting layer (the maximum data size accepted by each SN in the assisting layer is limited to 21 KB in each round of transmission). The $l_{i,B}$ is randomly set from 800m to 1200 m, and the $l_{i,j}$ is randomly set from 100 m to 300 m. In addition, we assume that the transmit powers $P_{i,B}$ and $P_{i,j}$ are 150 mW and 60 mW, respectively, and the channel bandwidth and carrier frequency are 1 kHz and 20 kHz. For the data task of each SN in the optimization layer, we consider that the data size is 10 KB, the number of CPU cycles required to calculate and process the data is 30 M cycles, and the size of the processed result data is set as 0.1 KB. Moreover, the computing power of each SN in the optimization layer is assumed to be 10 Mcycles/s, and the simulation parameters are summarized in Table 2.

The convergence performance of the proposed E-DDTMD algorithm and centralized decision scheme (CDS) [35] is demonstrated in Figure 4. In Figure 4, it can be seen that in the first 10 iterations, the curve of the proposed E-DDTMD algorithm rapidly drops close to the CDS. Until the 25th iteration, the algorithm converges to the CDS and then stabilizes. This shows that the proposed distributed algorithm has better convergence performance.

In Figure 5, we compare the total overhead of ’The Optimal Mechanism’ with the total overhead of ‘TAC’ and ‘TDBS’. It can be observed that the total overhead increases with the number of SNs in the optimization layer in the figure, because increasing the number of SNs in the optimization layer will increase the total data size accordingly. Compared with the two mechanisms of ‘TAC’ and “TDBS”, the total overhead of ‘The Optimal Mechanism’ increases more slowly, which proves that ’The Optimal Mechanism’ can effectively reduce the overhead of the optimization layer. (The number of SNs selected for mode *a*, *b*, *c*, and *d* are 2, 7, 4, and 5, respectively). 

Figure 6 compares the results of ’The Optimal Mechanism’ and the two mechanisms, ‘TAC’ and ‘TDBS’, when the size of each SN’s data size in the optimization layer varies. In this figure, the number of SNs in the optimization layer is set to 15. As shown in the figure, as each SN’s data size increases, the overhead of ‘TDBS’ increases linearly. The overhead of ‘The Optimal Mechanism’ and ‘TAC’ increases slightly (caused by the corresponding increase in the data size after calculating and processing the results), because when the amount of data increases, the overhead of processing data in mode *a* and mode *b* increases significantly, so ‘The Optimal Mechanism’ will choose mode *c* and mode *d* more.

Figure 7 shows the total energy consumption change of the SNs in the optimization layer when the computing power of each SN in the optimization layer changes, comparing ’The Optimal Mechanism’ with the two mechanisms of ‘TAC’ and ‘TDBS’. The number of SNs in the optimization layer is set to 15. As shown in the figure, with the increase of SN’s computing power in the optimization layer, the overhead of ‘TAC’ increases quadratically, which indicates that although increasing the local computing power of SNs can make them process more computing tasks locally and reduce the delay of calculation, it will also increase a lot of energy consumption. At the same time, the overhead of ‘TDBS’ remains unchanged, and the overhead of ’The Optimal Mechanism’ gradually increases until the computing power increases to 13 M cycles/s. This is because when the computing power of each SN increases to 13 M cycles/s, the optimization layer SN will not choose mode *c* and mode *d* to transmit data. Further, the overhead of SN selecting mode *a* and mode *b* does not change with the increase of computing power. At the same time, the simulation results show that keeping the computing power of SN at a relatively appropriate value is beneficial to control the energy consumption of SN.

In Figure 8, we compare adding a payment mechanism (Add-PM) and not adding a payment mechanism (Non-PM), showing that the addition of a payment mechanism effectively balances the energy consumption of SN in the optimization layer and the assistance layer. In particular, we assume that the assisting layer SN uniformly adopts mode *c* after receiving the data from the optimization layer SN, and the distance $l_{j,B}$ from the assisting layer SN j to the BS is randomly set from 600 m to 800 m. For the total energy consumption of optimization layer SNs, with the increase of the number of SNs in the optimization layer, the growth of Add-PM is slower than that of Non-PM, which indicates that the addition of a payment mechanism can effectively restrain the excessive dependence of optimization layer SNs on assistance layer SNs, and effectively balance the network energy consumption.

In Figure 9, in the assistance layer, when the data size that can be received by each SN is setting limit (setting) or not setting limit (Non-setting), compare the energy consumption variance of the assistance layer SNs (the assistance layer SNs adopt the same method c as above). Among them, compared with the ‘Non-setting’, the energy consumption variance of SNs in the assisting layer is much smaller in ‘Setting’, which proves that setting the limit on the received data size can effectively balance the energy consumption among SNs in the assisting layer. At the same time, when the number of SNs in the optimization layer increases to 30, the variance of ‘Setting’ tends to 0, because all 6 SNs in the assistance layer reach the upper limit of the data size that can be received.

## 8. Conclusions

In this paper, we first propose a novel HUWST framework. We then propose a game-based, energy-efficient underwater communication mechanism in the presented HUWST. It improves the energy efficiency of the underwater sensors personalized according to the various water depth layers of sensor locations. In particular, we integrate the economic game theory in our mechanism to trade off variation in communication energy consumption due to sensors in different water depth layers. Mathematically, the optimal mechanism is formulated as a complex NIP problem. A new E-DDTMD based on ADMM is thus further proposed to tackle this sophisticated NIP problem. The systematic simulation results demonstrate the effectiveness of our mechanism in improving the energy efficiency of UWSNs. Moreover, our present algorithm achieves significantly superior performance to the baseline schemes. In our proposed HUWST framework, if we want to use the proposed method, the global information in the network needs to be known. Therefore, we have to rely on the localization algorithm of an underwater sensor network [36]. In future work, we will consider the security of sensor nodes, as well as the reliability of transmission, which may involve semantic communication technology and blockchain technology.

## Figures and Tables

**Figure 2 sensors-23-05759-f002:**
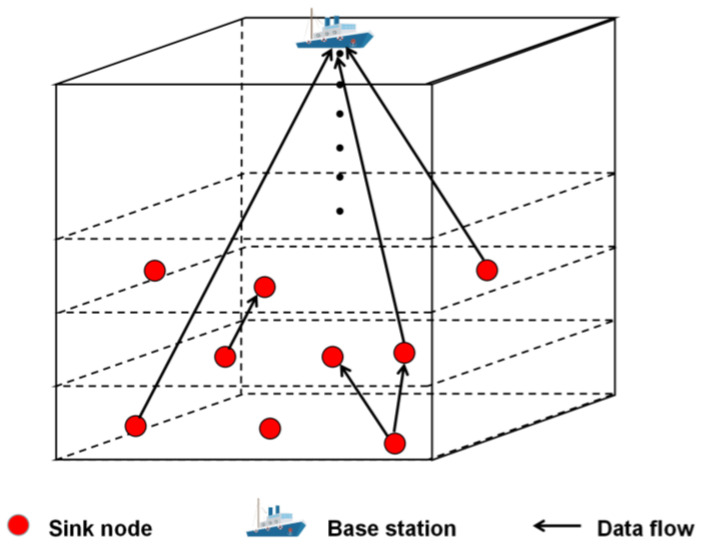
The SNs transmission model in the considered HUWST framework.

**Figure 3 sensors-23-05759-f003:**
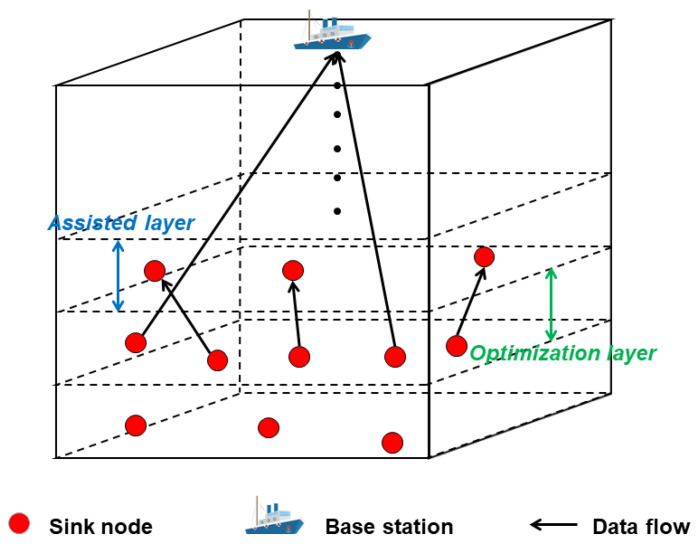
Three-layer transmission model.

**Figure 4 sensors-23-05759-f004:**
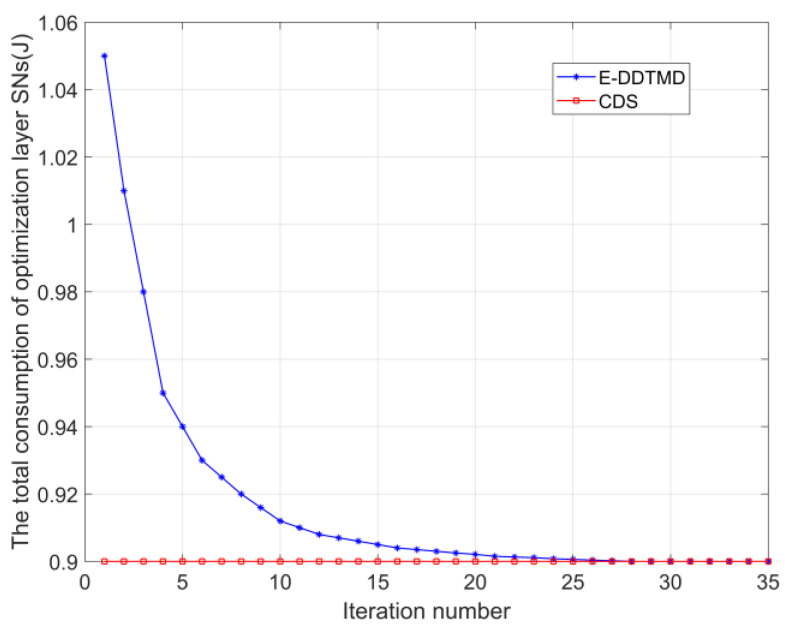
Convergence performance of the proposed E-DDTMD algorithm.

**Figure 5 sensors-23-05759-f005:**
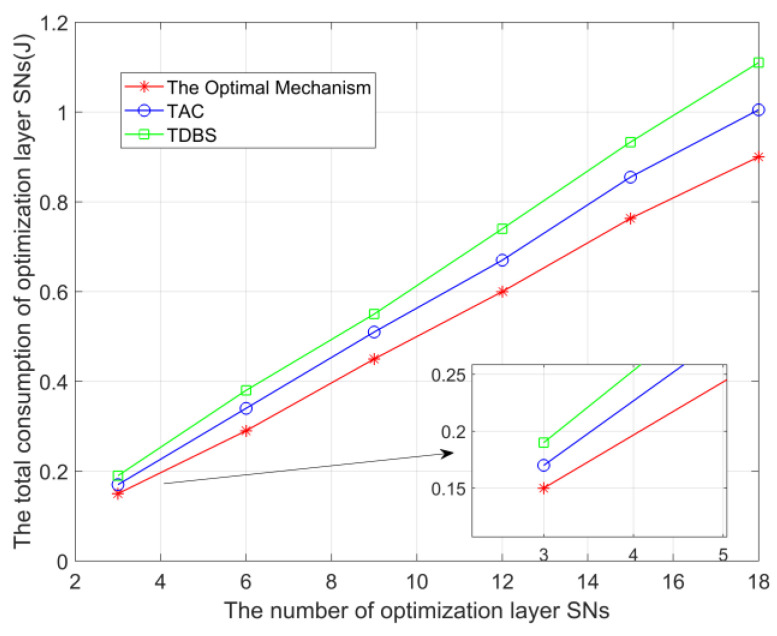
Comparison of total overhead when the number of SNs in the optimization layer is different.

**Figure 6 sensors-23-05759-f006:**
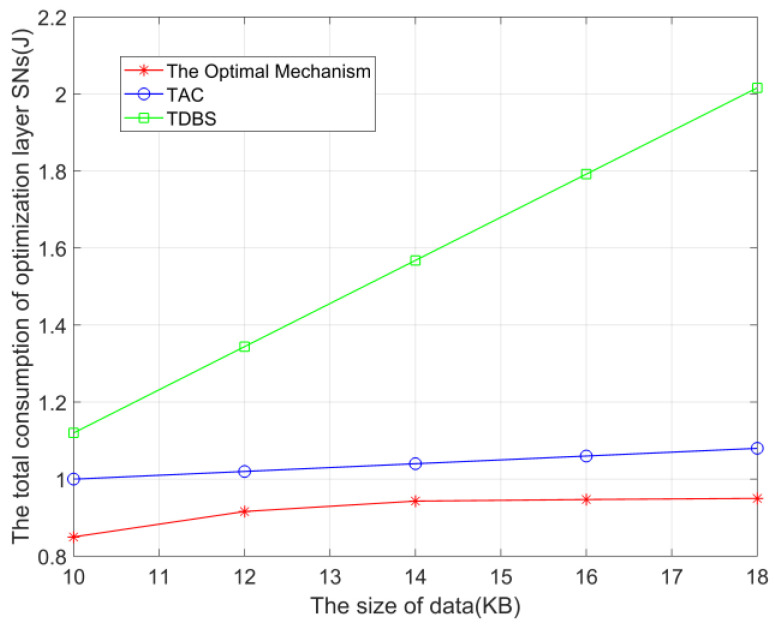
The total overhead when the amount of SN data in the optimization layer varies.

**Figure 7 sensors-23-05759-f007:**
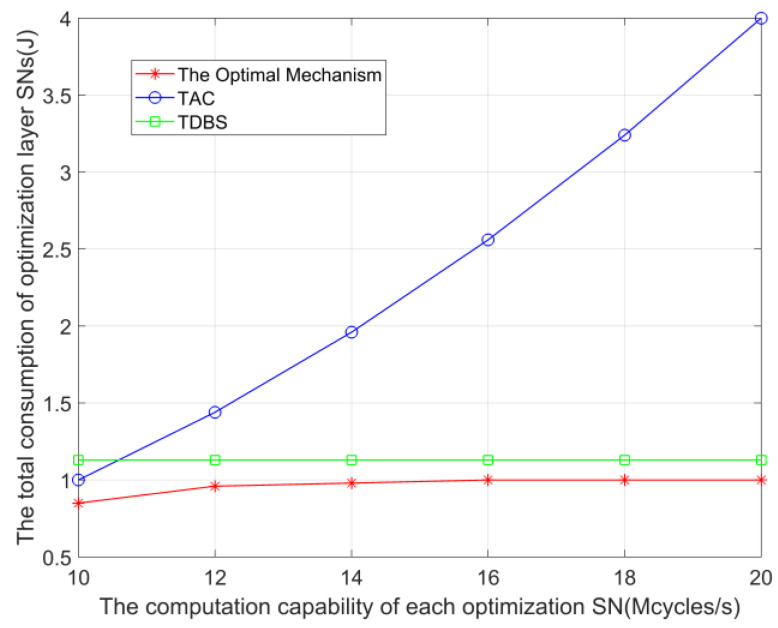
The total overhead when the computational power of the optimization layer SN varies.

**Figure 8 sensors-23-05759-f008:**
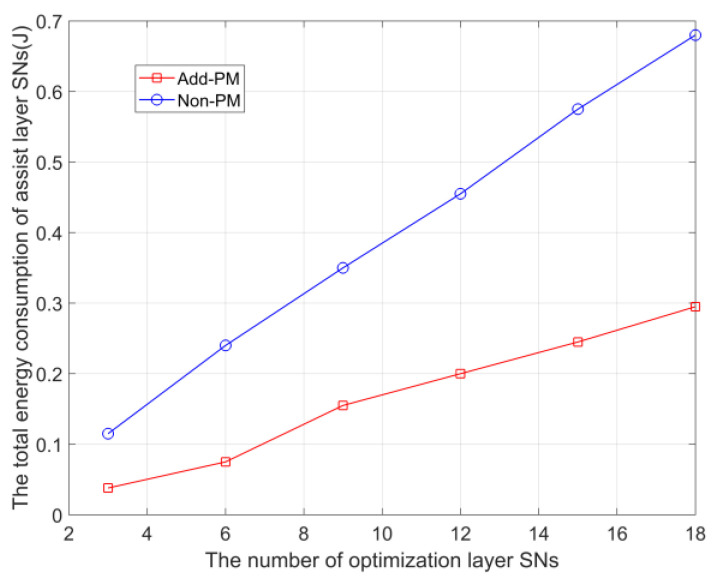
Comparison of upper SN energy consumption between Add-PM and Non-PM.

**Figure 9 sensors-23-05759-f009:**
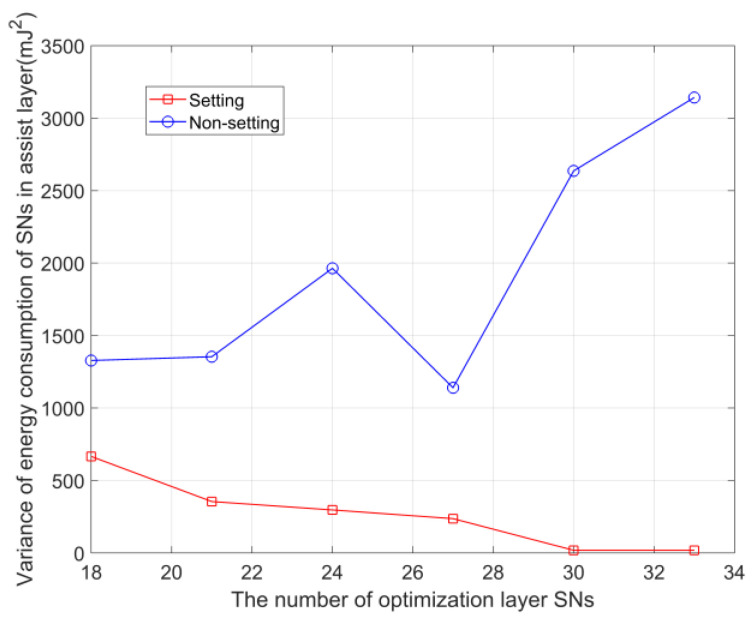
Comparison of SN energy consumption variance of the assisted layer with and without constraints.

**Table 1 sensors-23-05759-t001:** NOTATION.

Notation	Definition
$X_{i}$	The required CPU cycles of data $W_{i}$
D_i	The input data size of data $W_{i}$
$f_{i}^{L}$	The computation capability of SN i
a,b,d	The transmit decision vectors of SN i
$P_{i,B}$	The transmit power of SN i to BS
$P_{i,j}$	The transmit power of SN i to SN j
$B_{i}$	The available spectrum bandwidth of SN i
$f_{i}$	The carrier frequency of SN i
$D_{i}^{L}$	The data size of data $W_{i}$ after calculation
$l_{i,B}$	The distance between the SN i and BS
$l_{i,j}$	The distance between the SN i and SN j
$Z_{j}$	The maximum data size of SN j
R	The transmit rate

**Table 2 sensors-23-05759-t002:** Parameters.

Parameters	Definition
Bandwidth ($B_{i}$)	1 kHz
Carrier frequency ($f_{i}$)	20 kHz
Transmit power to BS ($P_{i,B}$)	150 mW
Transmit power to assisted layer SN ($P_{i,j}$)	60 mW
Size of computation task ($D_{i}$)	10 KB
Required CPU cycles of computation task ($X_{i}$)	30 Mcycles/s
Computation capability to each SN ($f_{i}^{L}$)	10 Mcycles/s
Size of the calculated data ($D_{i}^{L}$) The delay constraint ($T_{i}$)	0.1 KB 2 s

## Data Availability

The data used in this paper can be obtained by contacting the first author.

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
