# Peer review of "Energy-Efficient Data Transmission for Underwater Wireless Sensor Networks: A Novel Hierarchical Underwater Wireless Sensor Transmission Framework"

_sensors, 2023, doi:10.3390/s23125759_

Round 1

Reviewer 1 Report

Although the author have put much effort into this paper, however, the reviewer still has some reservations highlighted below:

1) The authors proposed a Energy-Efficient Data Transmission for Underwater Wireless Sensor Networks: a novel hierarchical underwater wireless sensor transmission framework , however, the presented work lacks novelty.

2)The overall contributions are not clearly presented. In the introduction, the authors simply presented what existing works have done and what they have done. However, the main differences from existing works should also be articulated. 

3) The author's contributions in the light of enlisting work are unclear, making it difficult to understand the advancements made to improve the state-of-the-art literature.

4) The terminology is not explained accurately, and some formula parameters are misdescribed in this paper. Also, the grammar in the paper is not complete, including the formula or the sentence.

5) The figures are not clearly stated, which makes it difficult to understand the underlying idea. For instance, Fig. 5, Fig. 6, and Fig.7 are not clearly elaborated.

6) Effective comparison of the presented work with literature and state-of-the-art is required.

7) The novelty seems to be limited, especially from a technical point of view. Parameters in Table 2 are not clear.

8)The terminology is not explained accurately, and some parameters are misdescribed in this paper. Also, the grammar in the paper is not complete, including the formula or the sentence. 

Reviewer 2 Report

See attached file.

Reviewer 3 Report

This manuscript proposed a HUSWT framework and in response to this framework, algorithms to reduce the energy consumption of underwater networks are proposed. However, as far as the reviewer being concerned, the content presented in this manuscript has little practical application. The reviewer even questioned whether the author of this manuscript had gone through real underwater acoustic(UWC) network experiments. The comments on this manuscript are as follows:

(1) The HUWST framework is not suitable for practical applications in shallow water UWC networks, and for deep-sea acoustic networks, clustering with reliable acoustic paths is nothing new.

(2) How is formula (3) derived? If it is based on Shannon's information theory, then whether the UWC channel that changes with time and space is applicable, it is necessary to know that Shannon's information theory is oriented to the Gaussian channel. What if Equation (3) doesn't hold? How can we talk about the optimization algorithm that follows?

(3) How does the optimization algorithm consider intercluster interference?

(4) UWC networks have been developed for many years, and it is no longer appropriate to verify them with simulation analysis. Even simulation analysis should fully consider the physical characteristics of the actual UWC channel and marine dynamic processes, although this is difficult to achieve in simulation analysis.

(5) There are some grammatical and spelling errors in the manuscript

Reviewer 4 Report

The paper proposes a novel hierarchical underwater wireless sensor transmission (HUWST) framework- a planned sensor node management method—and integrates economic game theory into the HUWST framework to trade off variations in communication energy consumption due to sensors in different water depth layers. Finally, an E-DDTMD Algorithm based on ADMM is further proposed to tackle this complicated NIP problem. The paper introduces this method from architecture construction, algorithm processing, and simulation verification. The article structure is reasonable, and the content is substantial. The simulation results demonstrate the mechanism's effectiveness in improving the energy efficiency of UWSNs. It's precious. However, some problems need to be explored and supplemented:

1. E-DDTMD Algorithm can improve transmission energy efficiency very well, but how about its performance on transmission time efficiency? 

2. Many word errors in the article need to be corrected, such as lines 68, 101, 532, etc.

Reviewer 5 Report

This manuscript proposes propose a novel hierarchical underwater wireless sensor transmission (HUWST) framework. However, there are more problems with the writing of this manuscript, and I recommend a major revision before acceptance. Here are some comments.

1.      It is noted that the manuscript needs careful editing by someone with expertise in technical English editing paying particular attention to English grammar, spelling, and sentence structure so that the goals and results of the study are clear to the reader.

2.      Please check some symbols and typesetting, such as “energy-efficient data transmission,” in line 28, “I SN in this layer” in line 166, “J SN” in line 168, “when??0 > ??1” and so on.

3.      Please check the equation (2), What’s the meaning of “10 ??? ? (?)”?

4.      In line 238 and 245, “the assistance of SN j ??i,?” and “Total cost of mode c ???” is strange. Please check for similar write-ups.

5.      In line 271, “? = {??}?ℐ,??”, so where is j? There are a large number of equations and symbols, please check carefully.

6.      Please check “Algorithm2” for the line number.

7.      The CDS is from 2017, is there an updated method used to evaluate the performance of the method in this manuscript?

Round 2

Reviewer 1 Report

The authors have put much efforts in revising the manuscript. The reviewer have no more concerns.

Reviewer 2 Report

All of my comments were properly addressed. Well done!
